# Manual Lymph Drainage for Post-COVID-19 Related Cough, Breathlessness, and Fatigue; Two Case Reports

**DOI:** 10.3390/healthcare11233085

**Published:** 2023-12-01

**Authors:** Bronwyn Overall, Kaori Langley, Janet Douglass

**Affiliations:** 1Overall Massage Therapy, Bucasia, Mackay, QLD 4750, Australia; bronnie66@bigpond.com; 2Kaori Langley Massage Therapy, Kangaroo Point, Brisbane, QLD 4169, Australia; kaori@kaorilangley.com.au; 3College of Public Health, Medical and Veterinary Sciences, James Cook University, Townsville, QLD 4810, Australia

**Keywords:** manual lymph drainage, long COVID, cough, breathlessness, fatigue, therapeutic intervention

## Abstract

Background: Persistent symptoms after SARS CoV-2 infection such as fatigue, shortness of breath, and cognitive dysfunction that cannot be explained by an alternative diagnosis have been termed long COVID and present a significant emerging public health problem. Current approaches include rehabilitation and symptom management involving multiple health disciplines and as yet there are no pharmaceutical approaches other than routine symptom management. Manual lymph drainage (MLD) has been used to support recovery during pulmonary rehabilitation and reduce chronic inflammation including symptoms associated with long COVID. Case description and outcomes: Two adult females who had reported long-COVID symptoms more than 10 weeks after the resolution of the acute infection were treated with MLD by Remedial Therapists trained in the Dr Vodder method of MLD. Respiratory function (Peak Flow Meter) and blood oxygen levels (Oximeter) were recorded before and after a one-minute sit-to-stand test prior to the treatment. The Dyspnea-12 Questionnaire, the Revised Piper Fatigue Scale, and Likert scales were used to collect client-reported outcomes. Six 45-min treatments were applied weekly, with a follow-up review and treatment at three months. In both cases, all outcomes improved after the third treatment with further improvement noted at three months. Conclusions: MLD may offer a non-invasive, non-pharmaceutical approach to the resolution of long-COVID symptoms such as cough, breathlessness, and fatigue.

## 1. Introduction

Global infection with SARS CoV-2 exceeds 650 million people with cases increasing by 250–650,000 per day [1]. A large proportion of people will go on to experience long-term symptoms such as breathlessness and fatigue [2,3], reducing functional capacity and presenting an enormous future risk to public health and disability services. Labelled long COVID, diagnostic guidelines are still unclear, however, the generally accepted criteria are post-infection symptoms that continue for 4–12 weeks and cannot be explained by an alternative diagnosis. Symptoms may begin after a period of recovery from the acute episode or persist from the initial illness, and may fluctuate or relapse over time [2].

While the long-term effects of SARS-CoV-2 infection remain largely unknown, there are numerous reports on the adverse impact of persistent fatigue, shortness of breath, and cognitive dysfunction on the everyday functioning of affected individuals [3,4]. The most common symptom reported is breathlessness, and cardiopulmonary studies have shown significantly lower main peak VO2 among people experiencing long COVID compared to asymptomatic subjects (25.8 ± 8.1 mL/min/kg vs. 28.8 ± 9.6 mL/min/kg, *p* = 0.017) [5]. Shortness of breath (61%), fatigue (54%), and cognitive problems (47%) are among the most frequently reported symptoms [6]. A large cohort study on over 800 people in Brazil who had been hospitalised for at least one day due to COVID-19 found that 6 months after discharge, 71% of survivors (*n* = 567) were experiencing limitations in their daily activities, and 6% (*n* = 45) rated this as severe. Pain and discomfort (65%), breathlessness (65%), and anxiety and depression (57%) were also reported among this cohort [7]. Abnormal mucus production, systemic inflammation, lung fibrosis, and post-viral inflammation have also been observed [4,8,9]. High levels of inflammatory mediators and activated immune cells remaining in the affected organs and tissues may account for, or contribute to long-COVID symptoms.

Proper functioning of the lymphatic system is essential in all wound healing and disease recovery and offers a novel target for the treatment of long-COVID symptoms. Initial lymph vessels are found in all loose connective tissue where they remove inflammatory mediators and pathogenic material from the interstitial spaces. Non-invasive manual lymph drainage (MLD) techniques are considered a core therapy in lymphoedema management [10] and are used in the symptomatic management of multiple chronic and inflammatory conditions [11]. The Dr Vodder method of MLD has been shown to increase lymph flow, affect autonomic tone (sympatholytic) [12], reduce systemic inflammation [13], and alleviate pain [14]. Usually applied to the skin and superficial fascia, MLD can be applied to vessels and organs in the deep system (under the deep fascia), including the organs of respiration and the central nervous system. Intensive drainage of the intercostal lymph pathways using MLD can assist in recovery from respiratory diseases such as bronchitis and pneumonia [15], and drainage of the soft palate is used for cerebral oedema and neuroinflammation [16].

The first report on the effect of lymphatic drainage on long-COVID-related fatigue was among 20 adults who received weekly treatments using the Perrin Technique, described as an effleurage movement and other manual articulatory techniques [17]. Patient-recorded symptom severity scores showed a decrease in physical fatigue, post-exertional malaise, a reduction in mental fatigue, back pain, and depression, as well as increased concentration and energy, and improved sleep after 9–12 weeks [17]. The authors conclude that MLD may benefit long-COVID patients and suggest that early intervention and supportive treatment at the end of the acute phase of infection may lessen acute phase symptoms and prevent them from becoming chronic. Australian Vodder Therapists have also noted good results (anecdotal data) using MLD for symptoms of long COVID.

Herewith we present two case studies on the effect of MLD in the resolution of long-COVID-related breathlessness and fatigue. Clinically relevant improvements were achieved after the third treatment in both cases and further improvement was noted at the follow-up review. Positive outcomes noted in these cases suggest MLD may be an effective, non-invasive, non-pharmaceutical, early intervention for long COVID.

## 2. Case Presentation, Treatment, and Outcome

Both patients were females aged 50 years who had tested positive for COVID-19 more than 10 weeks previously (Table 1). While the acute infection had resolved, both patients had persistent respiratory symptoms that were not present before the infection and complained of fatigue and reduced activity levels that were unusual for them. The past medical histories were unremarkable with no significant co-morbidities. Both patients were healthy and without the use of prescription medication before infection with COVID-19, however, both had required ongoing prescription medication for post-infection symptoms. The therapists were also females in their 50s and had been trained in Dr Vodder’s MLD for more than five years (Table 1).

## 3. Clinical Assessments

Symptom severity was recorded using both objective and subjective assessments. Therapists recorded peak flow and blood oxygen levels at every treatment before and after a one-minute sit-to-stand test and prior to commencing MLD. Patients completed the symptom self-assessment scales before the first treatment and before the follow-up review treatment.

There were three physical outcomes measured by the therapist prior to each treatment.
The Peak Flow Meter (Phillips Respironics, Farnborough, UK) measures peak expiratory flow from the lungs. The device is Therapeutic Goods Administration (TGA) approved and endorsed by Asthma Australia [18]. The best of three attempts at the hardest fastest possible breath into the measuring tube was recorded;The Heart Sure Pulse Oximeter (Aeon Technology, Victoria, Australia) is a digital device with a small cuff that slides onto a finger and detects heart rate and blood oxygen levels [19];For the sit-to-stand test, the subject repeatedly sits down and stands up from a straight-backed chair without using their arms (arms folded across the torso). The number of ‘stands’ completed in one minute was recorded.

There were three symptom questionnaires completed by the patient before the first treatment and before the follow-up review treatment.
The Dyspnea-12 Questionnaire [20]: the patient rated twelve statements on physical and affective problems related to breathing as none, mild, moderate, or severe;The Revised Piper Fatigue Scale [21], the patient scored 22 questions on unusual symptoms of tiredness and fatigue on a numeric scale between zero and ten, where ten was the most severe;Other self-identified symptoms of tiredness, fatigue, coughing, breathlessness, pain, wellbeing, asthma symptoms, and inactivity were collected on Likert scales from one to six where six was the most severe.

All treatments were conducted between June and September 2022. In both cases, weekly treatments were planned with a follow-up treatment and review at 10 weeks. After the initial consultation and first treatment, Patient A was unable to attend for three weeks due to sickness. After the second treatment, Patient B had a break of one week due to travel (Figure 1).

## 4. Therapeutic Intervention

Manual lymph drainage as described by Dr Vodder is the “gentle manipulation of the skin to generate directional stretch and shear forces into the underlying tissue and around the lymph vessels [15]”. The primary technique is a ‘stationary circle’ in which the therapist’s fingers are oriented flat to the skin without applying any downward pressure. During an active ‘stretch’ phase, the therapist gently moves the skin in a forward and circular path parallel to the skin. As the circle is completed the fingers become passive and the elastic recoil of the skin brings the tissue and therapist’s hand back to the original start position. Therefore, although the fingers and skin have described a circular path and delivered the required stretch and shear forces into the tissue, the location of the fingers on the skin has not changed. Five stationary circles are usually repeated in each position and timed to mimic the maximum rate of pumping in the lymph vessels (27–30 times per minute) [15]. For treatment of the face, stationary circles are used on the skin to relieve sinus congestion and for drainage of the intercostal lymph pathways; they are applied in the vertical plane by aligning the distal finger pads within a single intercostal space. The rotary technique is a whole-hand movement used to activate broad areas of drainage in the superficial lymph vessels and is performed with an open thumb to maximise the area of contact on the skin. A ‘bronchitis technique’, in which the therapist actively stretches the lower rib cage medially and inferiorly during exhalation can be used to help clear the lower lobes of the lungs.

Patient A was treated by Bronwyn Overall (BO) and Patient B was treated by Kaori Langley (KL), who are both qualified in Dr Vodder’s Manual Lymph Drainage and Combined Decongestive Therapy (Certificate in MLD and CDT) and are members of the Australasian Lymphology Association, Accredited Lymphoedema Practitioner Program. Both Therapists applied the fundamental techniques and sequences as described by Vodder, including deep drainage of the intercostal pathways [15]. The manual component of each treatment lasted 45 min and was adapted on each occasion to the presentation of the patient on that day. All treatments followed the fundamental principles of lymph drainage, including stimulation of proximal pathways before active drainage of distal tissue, and treatment of superficial pathways before drainage of deeper structures. If the patient was coughing, too congested, or unable to lie face down, then the treatment was performed in the supine or side-lying positions according to the comfort of the patient.

The main features of each session were stationary circles at the neck and shoulders, rotary technique over the skin of the ribcage, intensive intercostal drainage, and stationary circles over the parasternal and paraspinal lymph nodes. Variations between patient treatments included additional treatment of the face for sinus congestion in Patient A. patient A was treated supine for the first two treatments after which she was able to receive treatment lying prone. Patient B was initially unable to receive the bronchitis technique as it made her cough too much. The first five treatments were applied in supine and side-lying positions to reduce coughing, and she coughed during positional changes in these sessions. For treatment six and seven she was able to lie prone and without coughing during positional changes. In the second session, additional stationary circles were performed around the shoulders as she had reported some soreness at the deltoid after the first treatment. Written informed consent for the publication of treatment details and the outcome measures was provided by both patients (Appendix A).

## 5. Outcomes

The CARE (Case Reporting) guidelines were used to guide the reporting of these case studies (Appendix A) [22].

### 5.1. Therapist-Assessed Outcomes

Both patients presented with excessive cough, which was exacerbated each time they changed position on the treatment table and initially limited the use of deep breathing and bronchitis techniques. Both Therapists noted tenderness in the intercostal spaces, as well as chest and back soft tissue stiffness due to coughing. Episodes of coughing were markedly reduced, and intercostal tenderness eased in both cases after three treatments, allowing a greater range of treatment positions and techniques to be used thereafter. Neither patient was still experiencing excess coughing or intercostal tenderness at the follow-up review treatment.

Small improvements were recorded in peak flow volumes, blood oxygen levels, and sit-to-stand scores in both patients over the treatment and review period (Figure 2).

### 5.2. Patient-Assessed Outcomes

Scores on the Dyspnoea-12 Questionnaire, Revised Piper Fatigue Scale, and Self-reported symptoms (Likert scales) all improved over the course of treatment (Figure 3).

By the third treatment, Patient A reported a reduction in sinus congestion and breathlessness and also less fatigue with increased bouts of energy. After the review treatment, Patient A reported an improvement in functional capacity and stated that she felt the best she had since before the initial COVID-19 infection and was not breathless anymore. The patient commented on her ability to wear masks more comfortably at work as her breathing was not restricted, and her chest no longer felt tight. Her energy and motivation levels had increased, and she was happy with her overall wellbeing. She continued to sleep well, and no longer took any blood pressure medication or pain medication for aching joints.

After the first treatment, Patient B experienced some soreness and tenderness at the upper chest, bilateral axilla, and deltoid, dull aches along the spine at the back, loosening at the chest, and increased urination. Pain and soreness at the axilla and deltoid had resolved after the fifth treatment. Two days after the second treatment, she felt unwell with increased coughing and phlegm and needed to use her asthma medication more. These symptoms had settled by the fourth day. By the follow-up review, Patient B no longer needed to use the Ventolin at the end of the day, which had been the primary outcome in seeking treatment.

Both Patients reported sleeping better after the MLD treatments (Table 2).

### 5.3. Intervention Adherence and Adverse Events

Although MLD is inherently a gentle technique, the effect on autonomic tone must be considered [12], and Patient B had an exacerbation of her asthma symptoms two days after the second treatment and which resolved in a few days. In both cases, the therapist had to adapt the application of MLD to accommodate limitations imposed by the comfortable positioning of the patient and to reduce coughing, but no other treatment-related issues were reported. Both patients had agreed to attend weekly treatment sessions, however, this was interrupted by illness for Patient A and travel commitments for Patient B, who also exhibited significant stress levels due to personal issues. At the follow-up review, both patients reported they were happy with their treatment outcomes and intended to return for treatment of any continued symptoms, and for overall wellbeing.

## 6. Discussion

There are no previous studies or case reports on the treatment of long-COVID symptoms using MLD (as described by Vodder). These case studies are also the first report on objective measures of the effects of MLD on long COVID. The changes in Peak Flow Meter and Oximeter scores showed gradual improvement for both patients but these changes were small and may have improved over the same time frame without treatment [23]. The Dyspnoea-12 Questionnaire showed a positive trend in the improvement of breathing-related issues, and at follow-up, both patients answered ‘none’ to ten of the twelve questions. We could find only one previous reference to the effect of MLD on dyspnoea in a report on females affected by breast cancer-related lymphoedema [24]. The EORTC QLQ C30 quality of life questionnaire was used and showed that participants who received MLD from a therapist had better improvement in dyspnoea scores than those who performed a self-treatment protocol at home (d = −4.6, 95% CI = −9.1 to −0.15, *p* = 0.04).

Prior to the intervention, both patients returned high scores (≥4) on the Revised Piper Fatigue Scale, an indication that professional intervention was needed. After a course of treatment with MLD, neither patient met this threshold, which is consistent with the report by Heald and colleagues [17] who saw an average reduction of 50% in fatigue-related states after an average of nine sessions using the Perrin method with self-massage and exercises at home [17]. The Heald et al. cohort also reported improvement in sleep, which was also remarked on by both of our patients at the follow-up review.

The innovation in our study was in using a well-established treatment modality to address an emerging chronic health issue. The proactive study design allowed the therapists to use the same outcome measures and questionnaires, and the therapists communicated regularly throughout the treatment period to align their assessment and techniques as much as possible. Neither therapist was employed in an institutional or academic setting, nor had any previous experience using the measurement devices; therefore, it is likely there will have been some variation in the administration of the devices and questionnaires. It is inevitable that there will be variation in the exact treatment of each patient, as the nature of MLD requires it to be applied responsively on a case-by-case basis. The study results are limited by the small number of participants and unintended interruptions to the regularity of the treatment sessions. A further limitation is the unknown contribution of the medications that had been required prior to treatment in the resolution of symptoms. However, since these had been used without a change in symptoms before the MLD intervention, it is likely that the addition of MLD was the catalyst for the resolution of symptoms, possibly as an addition to, or in synergy with the medication.

MLD has previously been shown to improve quality of life in chronic conditions including lymphoedema [24], chronic venous insufficiency [25,26], and fibromyalgia [27], and to positively affect the autonomic nervous system [12]. With a lack of mental clarity, inability to focus, and other cognitive issues prevalent among people affected by long COVID [28], the results of these two cases indicate there may be potential for MLD to improve these symptoms.

## 7. Conclusions

Two prospective case studies were conducted to explore the use of MLD on long COVID-related breathlessness and fatigue, and positive outcomes were reported by both participants. The main recommendations arising from our study are to plan for at least three treatments with further treatments scheduled according to need; to adapt the treatment to the presentation and symptoms of the individual at each treatment; and to follow the fundamental principles of MLD as described by Dr Vodder. People experiencing long-COVID symptoms may benefit from MLD, and this well-established and evidenced-based modality should be considered for further investigation in the non-pharmaceutical and early intervention and resolution of long-COVID symptoms such as breathlessness and fatigue.

## Figures and Tables

**Figure 1 healthcare-11-03085-f001:**
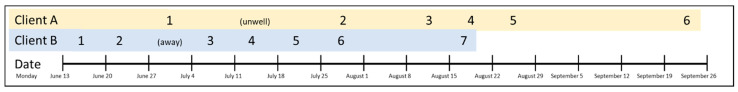
Timeline of treatment sessions. Treatments were conducted weekly between June and September 2022. Patient A was unable to attend for three weeks due to sickness and Patient B missed one week due to travel.

**Figure 2 healthcare-11-03085-f002:**
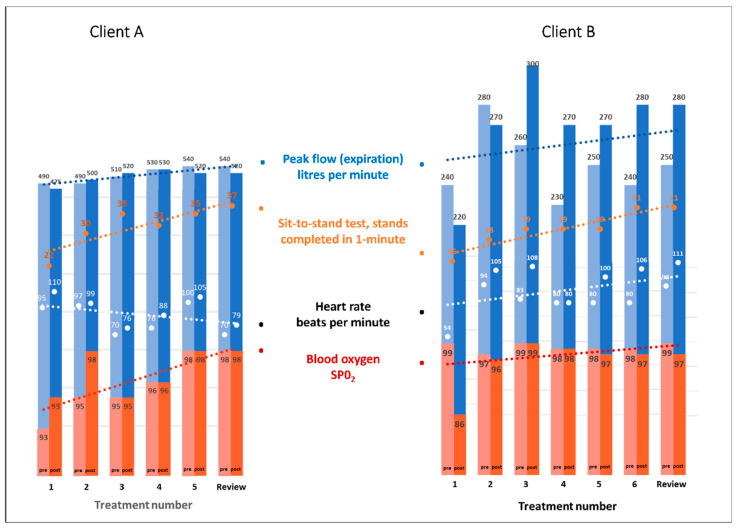
Scores for Peak Flow Meter, Oximeter, heart rate, and the 1-minute sit-to-stand test. Measures were taken before and after the sit-to-stand test and prior to the MLD treatment.

**Figure 3 healthcare-11-03085-f003:**
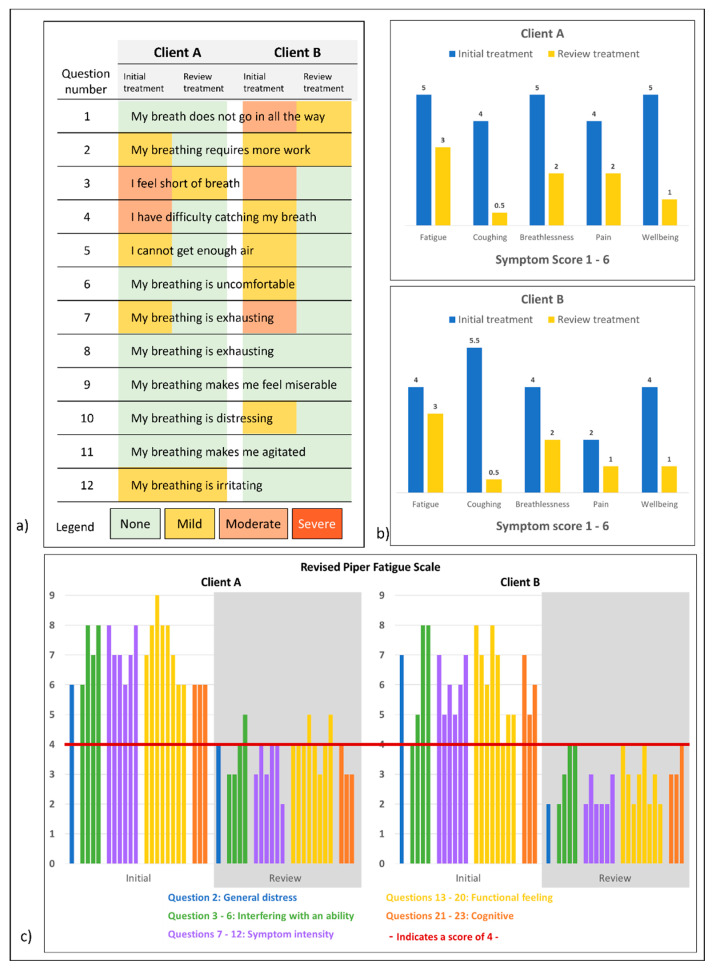
(**a**) Dyspnoea-12 Questionnaire, (**b**) self-reported symptoms (Likert scale), (**c**) patient scores on the Revised Piper Fatigue Scale. A score of ≥4 in multiple questions indicates the need for professional advice.

**Table 1 healthcare-11-03085-t001:** Participant characteristics, presenting symptoms, and therapist characteristics.

Participant	Case A	Case B
**Patient**		
*Age in years*	50	50
*Sex*	Female	Female
*Occupation*	Registered Nurse and Practice Manager (full-time)	Primary School Administration Officer (part-time)
*General health*	Good	Good
*Usual activity level*	Farmer/active	Fit/active/hiker
*Co-morbidities/medication*	None/none	None/none
*Time since +ve COVID-19 test*	11 weeks	15 weeks
*Medication required since infection*	BP Telmisartan 40 mg Mobic 15 mg	Symbicort inhaler and Ventolin for daily use
*Presenting concerns*	Tiredness Joint and back pain Breathlessness	Reduce asthma symptoms/medication
*Presenting symptoms*	Sinus congestion Headaches Fatigue Brain fog Joint pain (knees, ankles, wrists) Breathlessness Lack of social drive	Fatigue Asthma symptoms Weight gain (due to inactivity and comfort eating)
**Therapist**		
*Age*	56	50
*Sex*	Female	Female
*Substantive qualification*	Dip Remedial Massage	Dip Remedial Massage
*Years using MLD*	5 years since 2017	5 years since 2017

**Table 2 healthcare-11-03085-t002:** Revised Piper Fatigue Scale Q 24–28, free-text responses.

		Before the First Treatment	Before the Review Treatment
Overall, what do you believe is MOST directly contributing to or causing your fatigue?	Case A	Thought it was normal due to my workload, but it has not improved.	Work demands
	Case B	I don’t think my body is working properly.	Increase in energy from getting chest/breathing under control, so doing more activities which is making me tired.
Overall, the BEST thing you have found to relieve your fatigue is:	Case A	Typically, going to bed early, not socialising or doing anything too active over the weekend.	Good night sleep. Trying to let go of work. Gentle exercise. Relaxation. Deep breathing.
	Case B	Not get stressed about it, be mindful to do restful things to get through the day.	Not doing so much and taking a bit of time out.
Is there anything else you would like to add that would describe your fatigue better to us?	Case A	Usually really tired around 3–4 pm.	Slept exceptionally well after treatment and when using shakti mat.
	Case B	When I do a job or even driving in a car, I have to be very careful I don’t fall asleep.	It’s a feeling of being tired rather than being fatigued because I was unwell and struggle with breathing.
Are you experiencing any other symptoms right now?	Case A	Blurry vision when tired and headaches.	Not really.
	Case B	None given.	No—so very much improved.

## Data Availability

All data related to the case reports is included in the manuscript and Appendix A. Further data is not available due to privacy.

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
