# Peer review of "Manual Lymph Drainage for Post-COVID-19 Related Cough, Breathlessness, and Fatigue; Two Case Reports"

_healthcare, 2023, doi:10.3390/healthcare11233085_

Round 1

Reviewer 1 Report

Comments and Suggestions for Authors

What concerns me most about the present study are some methodological aspects that would require improvements that could perhaps be clarified or included.  

The article really seems interesting to me due to the need for assistance required by these patients and above all the need for research that manual lymphatic drainage requires or needs. 

As a main methodological error, I would like to point out that the physiotherapist who performs the intervention should be the same for treating both patients. In my opinion, this is already a major methodological error. 

Another aspect that I would like to highlight is that the medication taken by both patients could interfere with the result of the intervention, so I would recommend that this be stated in the text and, above all, make it clear whether they were taking it at the beginning of the intervention and for how long. 

It is also necessary to know and detail somewhat more clearly the times of data collection and for both patients. 

It makes it difficult to compare the results with only two subjects. 

Interruptions in the sessions could have conditioned the results. 

Since we are dealing with only two cases, we cannot make clear statements about the results obtained, although we can start with a way forward for future cases. 

The conclusions section should clearly respond to the objectives of the study. 

Author Response

Dear Reviewers and Editors,

We are very grateful for the time you have taken to assist us in improving this manuscript. This has guided the changes made and we trust you will now find the improvements are satisfactory.

The Authors.

Reviewer 1 Comments

What concerns me most about the present study are some methodological aspects that would require improvements that could perhaps be clarified or included. 

The article really seems interesting to me due to the need for assistance required by these patients and above all the need for research that manual lymphatic drainage requires or needs.

As a main methodological error, I would like to point out that the physiotherapist who performs the intervention should be the same for treating both patients. In my opinion, this is already a major methodological error.

Another aspect that I would like to highlight is that the medication taken by both patients could interfere with the result of the intervention, so I would recommend that this be stated in the text and, above all, make it clear whether they were taking it at the beginning of the intervention and for how long.

It is also necessary to know and detail somewhat more clearly the times of data collection and for both patients.

It makes it difficult to compare the results with only two subjects.

Interruptions in the sessions could have conditioned the results.

Since we are dealing with only two cases, we cannot make clear statements about the results obtained, although we can start with a way forward for future cases.

The conclusions section should clearly respond to the objectives of the study.

Author responses to Reviewer 1

Thank you for your considered remarks and valuable feedback on this manuscript. Responses to your comments are noted below and corresponding changes to the manuscript are given in blue highlights.

Clarity regarding methodology

These are separate case studies conducted by two therapists who had been trained in the same technique by the same teacher. It was considered that combining both studies in one manuscript would offer a more interesting and comprehensive report than by publishing two separate case studies. We agree that a prospective cohort study where all participants are treated by the same therapist would be preferable, however, we can only offer the two separate case studies here.

Medications

Medications had been used prior to the treatment phase and this has now been noted in the case presentation section with the addition of the phrase;

however both had required ongoing prescription medication for post-infection symptoms. The specific medications used are given in Table 1.

The following sentence has been added to the discussion to further highlight this limitation to the study results.

A further limitation is the unknown contribution of the medications that had been required prior to treatment in the resolution of symptoms. However, since these had been used without a change in symptoms before the MLD intervention it is likely that the addition of MLD was the catalyst for the resolution of symptoms, possibly as an addition to, or in synergy with the medication.

Clarification on data collection

The timing of the outcome measures has been clarified in the Clinical assessment section as follows;

There were three physical outcomes measured by the therapist prior to each treatment.

There were three symptom questionnaires completed by the client before the first treatment and before the follow-up review treatment.

Comparison of results, interruption to treatment

A statement on these limitations has been added to the Discussion and highlighted in yellow as both reviewers had commented on this aspect of the Discussion.

The study results are limited by the small number of participants and unintended in-terruptions to the regularity of the treatment sessions. A further limitation is the un-known contribution of the medications that had been required prior to treatment in the resolution of symptoms. However, since these had been used without a change in symptoms before the MLD intervention it is likely that the addition of MLD was the catalyst for the resolution of symptoms, possibly as an addition to, or in synergy with the medication..

Clarity of statements made on the results

We believe that we have now clarified the limitations to the results and have not made unsubstantiated claims on the effects of MLD in these cases. In particular the phrase …but these changes were small and may have improved over the same time frame without treatment. in the Discussion and revision to the final sentence in that section as follows have clarified this further.

…the results of these two cases indicate there may be potential for MLD to improve these symptoms.

Conclusion section

The following sentence has been added

Two prospective case studies were conducted to explore the use of MLD on long-COVID related breathlessness and fatigue, and positive outcomes were reported by both participants.

Reviewer 2 Report

Comments and Suggestions for Authors

Dear Editor and Authors,

It was quite interesting I must say reading the case report titled “Manual lymph drainage for post COVID-19 related cough, breathlessness, and fatigue; two case reports.” by Overall, Langley and Douglass from Australia.

Initially, the whole idea of utilizing lymphatic drainage massage to relieve symptoms of long Covid seemed preposterous!! Nevertheless the authors seem to support it by presenting a strong argument for manual drainage of lymph by showing clinical improvement in their two long-covid patients. I am also quite fond of attempts by independent researchers to present their work and experience!! The manuscript is clearly written in well understood language with only minor mistakes needing English editing. I do however have some minor commentary to offer. i.e.

Comments:

1.       The abstract needs some rewriting because it does not flow smoothly, the information offered is disjointed and there is no specific aim listed/presented!

2.       The introduction is a bit long and I would suggest shortening it and streamlining it!! For example a lot of the information about the anatomy and physiology of lymphatic drainage could be omitted or reduced/made more concise!!

3.       The aim/description of the case needs to be added at the end of the introduction! i.e. “Herewith we present two cases of long Covid patients….”  

4.       The case presentation section needs to be clearly marked! I suggest using a heading in bold!

5.       The case presentation needs also re-writing especially because the language used is not scientific!! i.e. we don’t use the term clients we say patients and also we report as female not women, we also don’t say both women were healthy with no health issues but their past medical history was unremarkable with no significant co-morbidities!!

6.       What is “brain fog”? Use the appropriate medical term please!

7.       What is Dip Remedial Massage? Do they mean Deep??

8.       The results and reporting by the patients (not clients – you are physiotherapists so they are patients for you too!!) is too colloquial and unscientific!! She said this, she reported that!! Make it more clinical, i.e. patient A reported an improvement in her functional capacity, ect

9.       We don’t need to know patient B personal life about her mother and her stress, ect. Make it more scientific!! Patient B exhibited significant stress levels due to personal issues!

10.   The discussion is short but well written. However, a limitation section were some of the issues reported in the results section could be moved to would be advisable.

In conclusion, this is an interesting report basically because it presents an unorthodox and novel application of a massage technique to improve long Covid symptoms. As a physician I pride myself of not been dogmatic and blinded by convention and dogma! Therefore, I feel this report should be published so that to bring to the scientific community an alternative therapeutic technique and initiate interest and exploration/experimentation – application.

However, the manuscript needs a good reworking before it is ready to be presented to the scientific community. Remember, you are attempting to publish in a medical journal and therefore you need to be able to report “in their language and jargon” in addition to (which I felt you have done well in that regard) report measurable variables and comparable assessments. I thank you for your time and effort and wishing you well.   

Comments on the Quality of English Language

Needs more scientific language to be used.

Author Response

Dear Reviewers and Editors,

We are very grateful for the time you have taken to assist us in improving this manuscript. This has guided the changes made and we trust you will now find the improvements are satisfactory.

The Authors.

Reviewer 2 Comments

Dear Editor and Authors,

It was quite interesting I must say reading the case report titled “Manual lymph drainage for post COVID-19 related cough, breathlessness, and fatigue; two case reports.” by Overall, Langley and Douglass from Australia.

Initially, the whole idea of utilizing lymphatic drainage massage to relieve symptoms of long Covid seemed preposterous!! Nevertheless the authors seem to support it by presenting a strong argument for manual drainage of lymph by showing clinical improvement in their two long-covid patients. I am also quite fond of attempts by independent researchers to present their work and experience!! The manuscript is clearly written in well understood language with only minor mistakes needing English editing. I do however have some minor commentary to offer. i.e.

Comments:

  1. The abstract needs some rewriting because it does not flow smoothly, the information offered is disjointed and there is no specific aim listed/presented!
  2. The introduction is a bit long and I would suggest shortening it and streamlining it!! For example a lot of the information about the anatomy and physiology of lymphatic drainage could be omitted or reduced/made more concise!!
  3. The aim/description of the case needs to be added at the end of the introduction! i.e. “Herewith we present two cases of long Covid patients….”
  4. The case presentation section needs to be clearly marked! I suggest using a heading in bold!
  5. The case presentation needs also re-writing especially because the language used is not scientific!! i.e. we don’t use the term clients we say patients and also we report as female not women, we also don’t say both women were healthy with no health issues but their past medical history was unremarkable with no significant co-morbidities!!
  6. What is “brain fog”? Use the appropriate medical term please!
  7. What is Dip Remedial Massage? Do they mean Deep??
  8. The results and reporting by the patients (not clients – you are physiotherapists so they are patients for you too!!) is too colloquial and unscientific!! She said this, she reported that!! Make it more clinical, i.e. patient A reported an improvement in her functional capacity, ect
  9. We don’t need to know patient B personal life about her mother and her stress, ect. Make it more scientific!! Patient B exhibited significant stress levels due to personal issues!
  10. The discussion is short but well written. However, a limitation section were some of the issues reported in the results section could be moved to would be advisable.

In conclusion, this is an interesting report basically because it presents an unorthodox and novel application of a massage technique to improve long Covid symptoms. As a physician I pride myself of not been dogmatic and blinded by convention and dogma! Therefore, I feel this report should be published so that to bring to the scientific community an alternative therapeutic technique and initiate interest and exploration/experimentation – application.

However, the manuscript needs a good reworking before it is ready to be presented to the scientific community. Remember, you are attempting to publish in a medical journal and therefore you need to be able to report “in their language and jargon” in addition to (which I felt you have done well in that regard) report measurable variables and comparable assessments. I thank you for your time and effort and wishing you well.  

Author responses to Reviewer 2

Thank you for taking the time to provide such comprehensive feedback, we greatly appreciate the opportunity this has afforded to improve the quality of the manuscript. Relevant changes have been highlighted in green.

Regarding the use of lymphatic drainage massage to relieve symptoms.

It is unfortunate that MLD is usually only considered in the treatment of lymphatic disorders and oedema. The vital role of proper tissue drainage in the healing of all injury and disease is generally not well understood, hence our emphasis on the anatomy and physiology of lymphatic drainage in the introduction (which has now been modified). As MLD Therapists we frequently use lymphatic drainage for a wide range of conditions including chronic sinusitis, recovery from pneumonia and symptom management in bronchitis and COPD.

  1. The abstract has been rewritten and we trust you will find it now flows better with a clear statement of aim.
  2. The introduction has been shortened and the description, with the role of the lymph system in healing and disease recovery replacing the previous A&P description as follows;

Proper functioning of the lymphatic system is essential in all wound healing and disease recovery and offers a novel target for the treatment of long-COVID symptoms. Initial lymph vessels are found in all loose connective tissue where they remove inflammatory mediators and pathogenic material from the interstitial spaces.

  1. The following statement has been added to the Introduction.

Herewith we present two case studies on the effect of MLD in the resolution of long-Covid symptoms.

  1. All section headings have been formatted with bold type.
  2. The term women has been replaced with female, and client has been replaced with ‘patient’ throughout, however please note the authors are Remedial Massage Therapists (not physiotherapists) and in Australia massage therapists are discouraged from using the term patient. The case presentation section has been rewritten, particularly the sentence concerning the past health history.

The past medical histories were unremarkable with no significant co-morbidities.

  1. The term ‘brain fog’ has been popularised in media reports on long-COVID and has been replaced with lack of mental clarity, and an inability to focus.
  2. Both therapists have a Diploma in Remedial Massage which is an Australian Qualification Framework Level 5 qualification. Physiotherapy is bachelor degree at AQF level 7.
  3. The results have been reviewed with the following phrase replacing the previous wording

After the review treatment, the patient reported an improvement in functional capacity

  1. Details of the stress circumstance for Pat4ien B have been replaced with the suggested phrase;

Patient B who also exhibited significant stress levels due to personal issues.

  1. The paragraph on limitations in the Discussion has been expanded and is highlighted in yellow as both reviewers had commented on this aspect of the Discussion.

The study results are limited by the small number of participants and unintended interruptions to the regularity of the treatment sessions. A further limitation is the unknown contribution of the medications that had been required prior to treatment in the resolution of symptoms. However, since these had been used without a change in symptoms before the MLD intervention it is likely that the addition of MLD was the catalyst for the resolution of symptoms, possibly as an addition to, or in synergy with the medication.

We are very grateful for your support and assistance with this manuscript. Massage Therapists are often involved in aftercare once conventional medical procedures have been completed, but it is very difficult and expensive for a non-academic practitioner operating in a small private practice to get their case studies published. Your support in bringing this alternative therapeutic technique to the scientific community is greatly appreciated.

Round 2

Reviewer 2 Report

Comments and Suggestions for Authors

Dear Editor and Authors,

Congradulations, you have implemented all suggestions this reviewer has made to a quite good degree and having read your manuscript once more I feel it is now much improved. As such I am happy to recommend its acceptance for publication. Good job.

Comments on the Quality of English Language

Language is fine - needs some minor editing at the proofing stage.